# Dose Reduction to Motor Structures in Adjuvant Fractionated Stereotactic Radiotherapy of Brain Metastases: nTMS-Derived DTI-Based Motor Fiber Tracking in Treatment Planning

**DOI:** 10.3390/cancers15010282

**Published:** 2022-12-31

**Authors:** Christian D. Diehl, Enrike Rosenkranz, Maximilian Schwendner, Martin Mißlbeck, Nico Sollmann, Sebastian Ille, Bernhard Meyer, Stephanie E. Combs, Sandro M. Krieg

**Affiliations:** 1Department of Radiation Oncology, School of Medicine, Klinikum rechts der Isar, Technical University of Munich (TUM), 81675 Munich, Germany; 2Institute of Radiation Medicine (IRM), Helmholtz Zentrum München, 85764 Neuherberg, Germany; 3Deutsches Konsortium für Translationale Krebsforschung (DKTK), DKTK Partner Site, 81675 Munich, Germany; 4Department of Neurosurgery, School of Medicine, Klinikum rechts der Isar, Technical University of Munich (TUM), 81675 Munich, Germany; 5Department of Diagnostic and Interventional Neuroradiology, School of Medicine, Klinikum rechts der Isar, Technical University of Munich (TUM), 81675 Munich, Germany; 6TUM-Neuroimaging Center, Klinikum rechts der Isar, Technical University of Munich (TUM), 81675 Munich, Germany; 7Department of Diagnostic and Interventional Radiology, University Hospital Ulm, 89081 Ulm, Germany

**Keywords:** brain metastases, stereotactic fractionated radiotherapy, fiber tracking

## Abstract

**Simple Summary:**

Management of brain metastases adjacent to the motor cortex and pyramidal tract bears an elevated risk for treatment-related morbidity and, hence, a reduced quality of life. For safe resection, navigated transcranial magnetic stimulation (nTMS) motor mapping combined with diffusion tensor imaging (DTI)-based fiber tracking (DTI-FT_mot.TMS_) is used to guide the neurosurgeon to preserve motor function. So far, during radiotherapy motor structures are not respected as organs at risk. In this study, the practicability of DTI-FT_mot.TMS_ in adjuvant radiotherapy planning of brain metastases was investigated, demonstrating a potentially significant dose reduction to cortical and subcortical motor structures.

**Abstract:**

Background: Resection of brain metastases (BM) close to motor structures is challenging for treatment. Navigated transcranial magnetic stimulation (nTMS) motor mapping, combined with diffusion tensor imaging (DTI)-based fiber tracking (DTI-FT_mot.TMS_), is a valuable tool in neurosurgery to preserve motor function. This study aimed to assess the practicability of DTI-FT_mot.TMS_ for local adjuvant radiotherapy (RT) planning of BM. Methods: Presurgically generated DTI-FT_mot.TMS_-based corticospinal tract (CST) reconstructions (FT_mot.TMS_) of 24 patients with 25 BM resected during later surgery were incorporated into the RT planning system. Completed fractionated stereotactic intensity-modulated RT (IMRT) plans were retrospectively analyzed and adapted to preserve FT_mot.TMS_. Results: In regular plans, mean dose (D_mean_) of complete FT_mot.TMS_ was 5.2 ± 2.4 Gy. Regarding planning risk volume (PRV-FT_TMS_) portions outside of the planning target volume (PTV) within the 17.5 Gy (50%) isodose line, the DTI-FT_mot.TMS_ D_mean_ was significantly reduced by 33.0% (range, 5.9–57.6%) from 23.4 ± 3.3 Gy to 15.9 ± 4.7 Gy (*p* < 0.001). There was no significant decline in the effective treatment dose, with PTV D_mean_ 35.6 ± 0.9 Gy vs. 36.0 ± 1.2 Gy (*p* = 0.063) after adaption. Conclusions: The DTI-FT_mot.TMS_-based CST reconstructions could be implemented in adjuvant IMRT planning of BM. A significant dose reduction regarding motor structures within critical dose levels seems possible.

## 1. Introduction

Brain metastases (BM) are the most frequent brain tumors, accounting for about 50% of all intracranial neoplasms, and the incidence is tenfold higher than that for gliomas [1]. Overall, 10–40% of all cancer patients will develop BM over the course of the disease and the incidence of BM is increasing, which results from better radio-oncologic and systemic treatment options for many cancer types, with improved survival and hence higher risk for BM during lifetime [2,3,4].

Large or symptomatic lesions can be disabling, thus requiring rapid resection for fast and persisting symptom relief [5,6]. However, especially BM close to the primary motor cortex and the corticospinal tract (CST) bears a high risk for treatment-related decline in motor function. Therefore, these lesions are challenging for treatment, underscoring the need for tools that can help to preserve motor function and reduce treatment-related morbidity. Lately, navigated transcranial magnetic stimulation (nTMS) has emerged as a valuable tool in neurosurgery [7,8]. With linking stimulation to three-dimensional (3D) neuronavigation, real-time motor mapping allows for an individual location of functional cortical motor spots that can differ from anatomical landmarks due to a tumor’s space-occupying effects, paired with potential functional reorganization [9,10,11,12]. In addition, nTMS motor mapping has been recently combined with diffusion tensor imaging fibertracking (DTI-FT_mot.TMS_), thus offering the possibility to trace the course of the individually mapped CST from the cortex to the medulla oblongata, in spatial relation to the lesion [13,14,15]. Therefore, preoperative DTI-FT_mot.TMS_ has been suggested to facilitate tumor resection while keeping motor function stable and, consequently, could improve clinical outcome [16,17,18].

Despite best medical treatment, local recurrence rates can still be up to 50% after removal of BM, which is related to remnant tumor cells necessitating adjuvant radiotherapy of the resection cavity to improve local control [19,20]. Although modern techniques are applied for adjuvant radiotherapy (RT), there are growing indications for radiation-mediated decline of motor function [21,22,23]. Routine sparing of the precentral gyrus and CST in RT planning is not supported so far, mainly because the real clinical impact is not sufficiently proven [24,25,26,27,28]. However, in the light of prolonged survival and better prognosis for cancer patients, preserving a patient’s quality of life and self-independence is of growing concern.

Against this background, this study investigated the implementation of DTI-FT_mot.TMS_ in adjuvant stereotactic RT of motor-eloquent BM for avoidance of motor structures.

## 2. Materials and Methods

### 2.1. Patient Selection

In this retrospective monocentric study, 24 patients with 25 BM resected during later surgery that were located close to the CST were included. All patients underwent tumor resection after having received nTMS motor mapping. After surgery, they were treated with hypo-fractionated stereotactic RT (hFSRT) to the surgical bed. Medical records were reviewed for demographic (sex, age, and grade-prognostic assessment (GPA) [29]), pathologic, oncologic treatment, RT planning, and clinical outcome data. Dosimetric data were extracted from the treatment planning system (TPS; Aria, version 13.0; Varian Medical Systems, Palo Alto, CA, USA; dose calculation AAA 13, dose grid spacing 2.5 mm). All patients were operated and irradiated at the academic medical centre in the period from June 2016 to October 2020.

### 2.2. Navigated Transcranial Magnetic Stimulation

Preoperative nTMS motor mapping was performed before tumor resection, based on a 3D contrast-enhanced T1-weighted sequence (repetition time (TR)/echo time (TE): 9/4 ms, 1 mm^3^ isovoxel covering the whole head) from presurgical cranial magnetic resonance imaging (MRI). These images were imported into the Nexstim eXimia NBS system (version 4.3. and 5.1.1; Nexstim Plc., Helsinki, Finland). The hemisphere ipsilateral to the metastases was motor mapped. Single pulse application with an intensity of 105% of the individual resting motor threshold (rMT) was used for stimulation of the cortical representations of the upper extremity, accordingly, for the lower extremity, a single pulse application with an intensity of least 130% of the rMT was applied [8,26,27]. A stimulation point evoking a motor-evoked potential (MEP) with an amplitude ≥50 µV and an MEP onset latency within the common limits for the upper or lower extremities was defined as a motor-positive spot [30,31,32].

### 2.3. Diffusion Tensor Imaging Fiber Tracking

DTI-FT_mot.TMS_ was generated based on the nTMS motor maps, applying a deterministic algorithm implemented in Brainlab Elements (version 3.1.0; Brainlab AG, Munich, Germany). Thus, presurgical MRI data including DTI (TR/TE: 5000/78 ms, voxel size of 2 × 2 × 2 mm^3^, one volume at b = 0 s/mm^2^, and 32 volumes at b = 1000 s/mm^2^), anatomical T2-weighted, and contrast-enhanced T1-weighted sequences were fused, and both the motor-positive spots and the ipsilateral brain stem were dedicated as regions of interest (ROIs) [8,13,33,34]. Fibers in between the ROIs (motor-positive nTMS spots and brain stem) with a minimum fiber length of 100 mm and passing a fractional anisotropy threshold of 0.1–0.2 were detected. A CST reconstruction originating from motor-eloquent areas as identified by nTMS was finally presented as a result from this process [8,13,33,34].

### 2.4. Elastic Image Fusion

For further data processing of the DTI-FT_mot.TMS_ results, preoperative and postoperative MRI data (with the same sequences) were fused. MRI-based elastic fusion (EF) was performed for the purpose of compensating for potential brain shift after tumor removal of brain metastases [35,36,37]. Correction of image distortions and shifts particularly for diffusion MRI were obtained using the EF algorithm (Brainlab AG, Munich, Germany). Selected structures as the presurgical mapped CST were considered concomitantly in this process [35,36,37].

Regarding this process, the edges of the metastasis based on the contrast-enhanced T1-weighted MRI-sequence were visually defined, and the spot closest to the DTI-FT_mot.TMS_ based CST was tagged. Linear measurements were then carried out from the tumor edge to the nearest CST fibers, and the distance was saved and defined as the minimum lesion-to-CST distance [14,15,38].

### 2.5. Target Delineation and Radiotherapy Planning

Presurgical MRI datasets with EF and DTI-FT_mot.TMS_ results of 24 patients who had previously undergone local adjuvant RT after BM resection were imported into the TPS (Eclipse, version 13.0; Varian Medical Systems, Palo Alto, CA, USA) for retrospective dosietric adaption. These images were fused with the original planning computed tomography (CT) scans using the automatic registration mode. In case of inaccuracy, manual registration was conducted.

The DTI-FT_mot.TMS_ results appeared as 3D objects. Pre-existing treatment plans were utilized. Target delineation was achieved as follows: the resection cavity including any residual tumor, in cases of subtotal resection and the surgical tract were defined as gross total volume (GTV), according to the valid consensus guideline [28]. Moreover, to cover microscopic spread, a margin of 2 mm was added to the GTV, thus delineating the clinical target volume (CTV) with adaptation to anatomical barriers such as the skull, falx, tentorium, and brain stem. Finally, in regard to patient movement, the CTV was expanded by 1 mm to cover minor setup errors, defining the representing planning target volume (PTV). The organs at risk (OARs, optical nerves, optical chiasm, brain stem, cochleae, lenses, lacrimal glands, bulbi, pituitary gland, and hippocampi) were delineated in line with an established consensus guideline [24]. Thereafter, the respective DTI-FT_mot.TMS_ of each patient was contoured as planning risk volume (PRV; FT_mot.TMS_) within the ARIA thresholding mode (range, 500- to 2000 Hounsfield units), with manual corrections. Mostly dual volumetric arc IMRT (VMAT) plans were calculated, with only one 3D plan applied. Dose prescription was 7 × 5 Gy, which was prescribed to the conformally enclosing 95%-isodose line. In two patients, original fractionation was 6 × 5 Gy, so plans were re-calculated to 7 × 5 Gy for better comparison. Plan normalization was homogenized for all plans, with 95% of the prescribed dose covering 99.0% of the target volume.

RT-mediated cortical atrophy and injury of white matter (WM) tracts appear dose-dependent beyond 20 to 30 Gy [39,40]. Subsequently, treatment plans were adapted by reducing the dose as much as reasonably achievable (spare) to FT_mot.TMS_ segments within the 17.5 Gy-isodose line, equivalent to 19.25 Gy in 2 Gy fractions with an alpha/beta of 3 (EQD2_3_). Portions of FT_mot.TMS_ within the PTV (overlap) were not spared, to ensure adequate treatment dose. All planning was performed by one experienced medical physicist to exclude inter-observer variability. In addition, the inverse planning process was strictly standardized, regarding constraints and structures used. In this retrospective analysis adapted plans were not used for real patient treatment.

### 2.6. Dosimetric Analysis and Comparative Evaluation

Spatial relation of FT_mot.TMS_ and the dose distribution was characterized by the proportional overlaps of FT_mot.TMS_ with specific isodose levels, as expressed in relative volume of FT_mot.TMS_ receiving at least 100% (35 Gy), 95% (33.25 Gy), 90% (31.5 Gy), 50% (17.5 Gy), and 25% (8.75 Gy) of the prescribed dose (V100%_35Gy_, V95%_33.25Gy_, V90%_31.5Gy_, V50%_17.5Gy_, and V25%_8.75Gy_), based on the dose-volume histogram (DVH). For both plans (FT_mot.TMS_ regular and FT_mot.TMS_ spared), DVH parameters for PTV, FT_mot.TMS_, and organs at risk (optical nerves, optic chiasm, brain stem, hippocampi, lacrimal glands, lenses, eye bulbs, cochleae, and pituitary gland) were evaluated for median dose (Dmean) and maximum dose (Dmax). Dose to healthy brain was assessed by the absolute volume of brain (minus cavity) receiving at least 28 Gy (V28Gy) in seven fractions, which is the equivalent biological effective dose (alpha/beta = 3) of 12 Gy (V12Gy) in single-fraction radiosurgery, or 24.4 Gy (V24.4Gy) for FSRT with 5 fractions [26,41].

Dose conformity was assessed based on the following parameters and formulas:

I. Coverage $=\frac{\mathrm{PTIV}}{\mathrm{PTV}}$ [42].

This parameter is applied to see how well the PTV is covered by the prescription dose, where PTIV is the target volume encompassed by the prescription isodose volume (PIV)(=PIV ∩ PTV). A value of/or close to 1.0 would indicate a perfect coverage.

II. Selectivity $=\frac{\mathrm{PTIV}}{\mathrm{PIV}}$ [42].

Selectivity is used as an indicator to see how ideal the prescription dose fits the PTV, thus covering areas beyond the PTV. In other words, higher values stand for less unnecessary coverage of normal brain beyond the PTV and hence better selectivity.

III. Conformity index CI_Paddick_ $=\frac{\mathrm{PTIV}\;\times \;\mathrm{PTIV}}{\mathrm{PTV}\;\times \;\mathrm{PIV}}$ [43].

Expressing coverage × selectivity, plan quality decreases with lower index values, with an ideal value being one.

Clinical motor function both preoperatively and postoperatively, on follow-ups, was analysed on retrospective chart review and expressed according to the British Medical Research Council scale (0–5) with 0/5 = no contraction, 1/5 = flicker/trace contraction, 2/5 = active movement (AM) with gravity eliminated, 3/5 = AM against gravity, 4/5 = AM against resistance, and 5/5 = full strength [44].

For assessment of local control, images were assessed in adaption of the response assessment criteria, proposed by the Response Assessment in Neuro-Oncology Brain Metastases (RANO-BM) group [45]. Radiographical response was measured multidimensionally and was recorded for every resection cavity postoperatively, at every follow-up visit. Lesions were considered measurable in case of nodular contrast enhancement visible on two or more axial slices and measurable with a minimum size of 5 mm (longest diameter (LD)). For measurable lesions >5 mm and <10 mm, an increase or decrease in LD of <3 mm was considered a stable disease (SD), and an increase of ≥3 mm compared to early postoperative MRI (within 48 h after resection) was considered a progressive disease (PD). Regarding measurable lesions of ≥10 mm, a minimum increase of 20% in LD was considered PD, otherwise SD.

### 2.7. Statistical Analyses

Statistical analyses and creation of graphs were conducted with SPSS (version 24.0; IBM Inc., Armonk, NY, USA). Descriptive statistics were applied for patient- and tumor-related characteristics as well as doses and volumes investigated in the present analysis.

Tests for statistical significance between values were performed using Pearson´s correlation analyses and *t*-tests for paired samples. Survival and local progression were estimated using the Kaplan–Meier function, and for local progression, patients dying without evidence of local recurrence, and patients remaining free of local recurrence at the end of follow-up were censored. The threshold of statistical significance was set at *p* < 0.05.

## 3. Results

### 3.1. Patient Characteristics

In this study, a total of 24 patients with 25 histo-pathologically proven BM close to the motor cortex and/or CST were analysed, which were resected during neurosurgery. The female-to-male ratio was 17:7 and the median age at diagnosis was 60.4 years (range, 32–78 years). Furthermore, non-small cell lung cancer (n = 7) and breast cancer (n = 5) were the most frequent primary tumors.

The left hemisphere was predominantly affected (16 vs. 8), 14 BM were in the frontal lobe and 7 in the parietal lobe. The mean graduated prognostic assessment (GPA) score was 2.2 ± 0.8 (range, 1–3.5) and 2.0 ± 0.9 (range, 0.5–3.5) pre- and postoperatively, respectively (*p* = 0.07). Mean volume of the resection cavity as measured on planning MRI was 7.1 mL ± 8.6 (range, 0.5–31 mL) and the volume of FT_mot.TMS_ was 54.6 mL ± 18.7 (range, 19.5–91.0 mL), respectively. The closest distance from the edge of the cavity to the CST (lesion-to-CST-distance) had a mean of 0.28 cm (range, 0–1.2 cm).

The PTV overlapped with FT_mot.TMS_, in 17 out of 25 cases, on average with 0.5% (range, 0–1.2%) of FT_mot.TMS_. Between FT_mot.TMS_ and PTV, the mean distance was 0.1 cm (range, 0–1.3 cm).

Summarized patient and tumor characteristics are displayed in Table 1.

### 3.2. FT_mot.TMS_ Dose Statistics

The intersection of FT_mot.TMS_ with the 100%, 95%, 90%, 50%, and 25%-isodose line (V100%, V95%, V90%, V50%, and V25%) was 1.7% (range, 0–7.1%), 2.4% (0–9.4%), 3.1% (0–11.4%), 10.9% (0.4–38.7%), and 22.0% (6.1–51.2%) of the FT_mot.TMS_, respectively. With sparing FT_mot.TMS_, V100%–V25% were reduced by −88.2% to 0.2% (0–1.1%, *p* = 0.01), −83.5 ± 12.1% to 0.5% (0–2%), −74.2% to 0.8% (0–2.7%), −66.8 ± 18.7% to 4.3% (0–12.8%), and 34.5% to 14.2% (2.2–30.2%, *p* < 0.01; Figure 1). There was no statistically significant change of FT_mot.TMS_ Dmax (34.7 ± 4.0 Gy vs. 33.6 ± 7.8 Gy, *p* = 0.18; Figure 1, Table 2).

In regular plans, FT_mot.TMS_ Dmean was 5.2 ± 2.4 Gy, with plan adaption FT_mot.TMS_ Dmean was reduced by 24.7% (range, 4.4–76.7%) to 3.8 ± 1.8 Gy (*p* < 0.001). For PRV-FT_TMS_ segments that were beyond the PTV (FT_mot_exPTV), reduction of Dmean was -26.8% (range, 4.4–76.7%, *p* < 0.01) from 4.8 ± 2.1 Gy to 3.3 ± 1.4 Gy (*p* < 0.001; Figure 2A). Within the 50% (17.5 Gy)-isodose level, the relative reduction of FT_mot.TMS_ (FT_mot_exPTV_50%_) was minus 33.0% (range, 5.9–57.6%) from 23.4 ± 3.3 Gy to 15.9 ± 4.7 Gy (Figure 2B). Regarding portions within the 25% (8.75 Gy)-isodose level (FT_mot_exPTV_25%_), there was a decrease of Dmean from 17.6 ± 4.2 Gy to 12.5 ± 3.6 Gy (−29.2 ± 12.2%; Figure 2C, Table 2). The was no significant correlation between relative reduction of FT_mot.TMS_ Dmean with the overlap of FT_mot.TMS_ and PTV (−0.14, *p* = 0.5), distance between FT_mot.TMS_ and cavity (0.29, *p* = 0.16), and the distance between FT_mot.TMS_ and PTV (0.34, *p* = 0.97). An example case with DVH is illustrated in Figure 3.

### 3.3. PTV Coverage, Conformity, and Outcome

With FT_mot.TMS_ avoidance, the PTV coverage did not significantly change, thus keeping an ideal value of 1.0 vs. 1.0 (*p* = 0.08). Prescription isodose volume (PIV_35Gy_) enlarged from 27.4 ± 24.4 mL to 28.0 ± 25.3 mL (*p* = 0.004), thus selectivity (0.88 ± 0.0 vs. 0.83 ± 0.1; *p* < 0.001) and Paddick-Conformity Index (0.88 ± 0.1 vs. 0.83 ± 0.1; *p* < 0.001) dropped accordingly. PTV Dmean did not significantly change with 35.6 ± 0.9 Gy vs. 36.0 ± 1.2 Gy (*p* = 0.064), both PTV Dmax and D2% (Gy) increased from 37.2 ± 0.9 Gy to 38.6 ± 1.2 Gy (*p* < 0.001) and from 37.0 ± 0.5 Gy to 37.7 ± 1.1 Gy (*p* < 0.001), respectively (Table 3).

Dose expositions of brain and OARs were assessed for comparison of both treatment plans. Dose to the healthy brain (brain—cavity) was estimated with absolute volume receiving 28 Gy or more (V28 Gy in ml). With FT sparing V28 Gy increased was 25.9 ± 15.9 mL compared to 25.3 ± 14.9 mL without FT avoidance (*p* = 0.4). All other OAR dose constraints were met and are detailed in Table 4.

### 3.4. Clinical Outcome Data and Follow-Up

Based on Kaplan–Mayer analysis, estimated local control (LC) and overall survival at 12 months after surgery were 85.9% (95%-CI 71.0–100%) and 68.1% (95%-CI 48.5–87.7%), respectively. Kaplan–Mayer curves are displayed in the Appendix A. One patient had a new grade 4/5 motor deficit postoperatively that subsided quickly until adjuvant RT. According to BM-RANO, there were two cases of suspected PD at 10 and 8 months after surgery, respectively: one patient died, secondary to extracranial tumor progression before any advanced imaging or biopsy could be performed, the other patient had salvage whole-brain radiotherapy (WBRT) for distant progression. A third patient had local PD, which was confirmed by 18-fluor-ethyl-positron-emission tomography. Based on retrospective analysis those patients did not reveal new motor deficits. All BM-RANO target lesions had growth towards the CST, but numbers were too low for deeper interpretation of growth patterns in relation to DTI-FT_mot.TMS_. One patient had symptomatic RN with predominantly progressive peri-cavitary edema, causing headaches and deterioration of pre-existing hemiparesis due to post-surgical hemorrhage. In this patient, RN was treated with 3 cycles of bevacizumab 7.5 mg per m^2^ body surface every three weeks, and symptoms subsided with regressive edema on follow-up scans.

## 4. Discussion

In this study, we implemented nTMS-based DTI-FT of the CST in postoperative fractionated stereotactic RT planning of motor-eloquent BM. Plan adaptions for sparing subcortical motor structures resulted in significantly lower dose expositions of motor fiber tracts in the high-dose area despite high dose gradients offered by stereotactic techniques.

Advances in systemic cancer management have improved survival for numerous types of solid cancer, therefore the number of patients harbouring BM is increasing [46,47]. Small clinical inapparent BM are commonly treated with stereotactic radiosurgery. According to established guidelines, resection should be carried out in patients with single BM and controlled primary disease or when histopathologic diagnosis is crucial for decision-making in cancer management, e.g., in cases of unknown primary tumors or when changes in molecular profiles compared to the primary tumor are suspected [48,49,50]. Patients with large and/or symptomatic lesions benefit from neurosurgical resection regarding rapid symptom relief, whereas a preferably extended resection is prognostically relevant for local control [51,52]. Of note, supra-marginal resection can even improve local control but with a higher risk of neurological deficits [53]. Therefore, in the management of motor-eloquent BM, advanced tools for mapping of cortical and subcortical structures, subserving motor function, are needed to reduce treatment related morbidity [54,55,56].

Therefore, functional pre-operative, nTMS based mapping of the motor cortex has been implemented in neurosurgery for sparing the individual primary cortical motor areas [44,57]. nTMS-based cortical mapping has been demonstrated to correspond with direct electrical stimulation (DES), which is regarded as the reference standard of intraoperative electrophysiological cortex mapping [18,58,59]. As further development, nTMS-generated cortical maps can be linked to DTI-FT, offering an individual function-based CST tractography, thus brain tumors adjacent the cortex and CST can be best resected, while motor function is preserved [13,14,15]. In contrast to peace-mal resection, en-bloc dissection along the brain-tumor interface is beneficial for local control, and decreases the risk for leptomeningeal spread, implying a relatively high risk for local recurrence through remnant cells, after resection [60,61,62]. Therefore, adjuvant RT is strongly needed to eliminate residual microscopical tumor spread in the resection cavity to prolong local control and improve outcome [19,20]. Several studies have demonstrated excellent local control rates up to 93% with hFSRT to the surgical bed [63,64,65,66]. Stereotactic radiation techniques enable a precise dose deposition to the cavity and the surrounding brain, with steep dose gradients to spare healthy brain tissue and OARs such as brain stem, optic chiasm, and optical nerves. Contrary to surgery, functional cortical and subcortical brain regions are not specifically spared and declared as structures at risk. Radiation-mediated injury of WM tracts such as the CST is seen as one cause of motor dysfunction after RT. Specifically, WM degeneration is mainly mediated by impaired oligodendrogenesis, vascular injury, and neuro-inflammation, which appear to happen in a linear dose-response pattern [67].

Generally, DTI quantifies spatial diffusion processes of water molecules, permitting conclusions on changes in WM microstructure [68,69]. Thus, DTI metrics such as mean diffusivity and fractional anisotropy can be used as biomarkers for RT-mediated injury of interhemispheric WM tracts and corpus callosum, associated with a decline in processing speed, or microstructural damage of cerebral motor structures, including the pyramidal tract, causing deterioration of fine motor skills [23,70].

So far, studies investigating dose exhibition of the motor cortex and CST in the setting of adjuvant stereotactic cavity RT after removal of BM are scarce. We therefore investigated 24 patients with 25 supratentorial BM for whom preoperative nTMS-based DTI-FT was retrospectively implemented into treatment planning for dose avoidance of the motor cortex and CST. In this cohort, Dmean of the PRV for motor structures (FT_mot.TMS_) was 5.2 ± 2.4 Gy (range, 1.8–12 Gy) with a mean cavity-to-tract distance of just 0.2 cm (range, 0–1.0 cm), and in 17 of 25 cases a proportional overlap of FT_mot.TMS_ with the PTV of 1.1 mL (range, 0.1–2.9 mL) was observed. For ensuring adequate treatment dose to the PTV, only FT_mot.TMS_ segments beyond the PTV were spared (FT_mot_exPTV). Thus, FT_mot_exPTV could be significantly reduced by −26.8% (*p* < 0.001) (Figure 2A). When only looking at portions within the “high-risk” isodose line (17.5 Gy) (FT_mot_exPTV_50%_), dose reduction was even more prominent with −33% (range, 5.9–57.6%) (Figure 2B). Accordingly, the volume of the FT_mot.TMS_ receiving the highest dose (equal to or more than 95% of the prescription dose (V95%)) dropped by −83.5 ± 12.1% (*p* < 0.001), respectively (Figure 1). Those numbers demonstrate the potential dose reduction to motor areas that can be achieved despite the yet steep dose gradients offered by stereotactic RT.

However, median treatment dose (Dmean) and dose coverage of the PTV were not negatively affected. During our procedure of plan adaptation, the prescription isodose volume (PIV_35Gy_) enlarged from 27.4 ± 24.4 mL to 28.0 ± 25.3 mL (*p* = 0.004) translating into less ideal, yet still appropriate numbers for selectivity (0.88 ± 0.0 vs. 0.83 ± 0.1; *p* < 0.001) and Paddick-Conformity Index (0.88 ± 0.1 vs. 0.83 ± 0.1; *p* < 0.001), with values of our present study being in the range of previous studies [43,71,72]. For better dosimetric comparison regarding FT_mot.TMS_, plans were not optimized in terms of individual normalization methods; yet we found higher values with plan normalization for 95% of dose covering 99.0% of the PTV, compared to covering 99.9%: Selectivity_spare_, 0.83 ± 0.1 vs. 0.76 ± 0.1 (*p* < 0.001), and Paddick-CI_spare_, 0.81 ± 0.1 vs. 0.76 ± 0.1 (*p* < 0.001).

Application of nTMS-based motor cortex mapping in hFSRT planning was investigated in a previous study reporting on 30 patients with resected BM, adjacent to the precentral gyrus, receiving adjuvant hFSRT, of whom 11 patients were eligible for plan optimization. The Dmean of the nTMS-based primary motor cortex was 23.0 Gy in 7 fractions, with cortical sparing a significant dose reduction of 18.1% to 18.9 Gy could be demonstrated, being obtained without affecting treatment doses to the PTV [73]. A further 19 patients exhibited low motor cortical dose exposition of 9.7 Gy and were not further analyzed. Another study investigated 22 patients receiving stereotactic RT to 23 cavities after removal of BM, and Dmean of the nTMS-defined motor cortex was 9.66 Gy, and after plan optimization for avoidance Dmean was 6.49 Gy [74]. Furthermore, DTI-FT_mot.TMS_ integrated in adjuvant RT planning of motor-eloquent high-grade glioma (HGG) has been recently demonstrated to enable significant dose reductions for avoidance of the CST [75]. So far, there are only a few studies demonstrating the potential impact of radiation to motor structures on motor function. In one study, DTI-FT_mot_ has been applied in treatment planning of Gamma Knife (GK) stereotactic radiosurgery (SRS) of solid tumors for pyramidal tract sparing, with 74 patients (BM 50%, mean volume 28.9 ± 26.2 mL) divided in two groups: DTI-avoidance (CST Dmax < 15 Gy) vs. control goup [75]. After 6 months, the DTI-avoidance group had significantly lower complication rates (not specified by authors), higher performance scores (Karnofsky Performance Status), and higher muscle strength of the affected limb [76]. Other authors propagated a dose constraint for the volume of the pyramidal tract receiving <20–25 Gy in GK-SRS, based on a cohort of 24 patients stereotactically treated for arterio-venous malformations (AVMs) [77]. They reported on two patients with large AVMs within the internal capsule having permanent motor complications, and for those patients, V25Gy and V20Gy were 8400 mL/302 mL and 400 mL/89 mL, respectively [77]. Generally, the internal capsule seems more predisposed for RT-induced WM injury than the corona radiata, most likely due to the higher fiber density [23,77]. Another recent trial demonstrated WM injury based on DTI metrics from fractionated RT doses > 20–30 Gy which were associated with a decline in motor functions [23]. The aforementioned studies used DTI-based tractography only, while DTI-FT_mot.TMS_ enabled operator-independent and function-based ROI seeding for CST reconstruction, linking functional cortical motor spots with corresponding CST delineation. Thus, there are no clear dose constraints in terms of RT-mediated WM degeneration and or RT-related functional motor dysfunction yet. RT induced damage of white matter tracts appears with doses higher than 20 Gy and 30 Gy, after 9–11 months and 4–6 months, respectively [39]. In regard to dose related cortical atrophy, the average reduction of cortical thickness increases with higher doses. A total of 34.6 Gy was demonstrated as an average break point dose with higher thinning per Gy, beyond that dose [40]. In our study on hFSRT for larger treating volumes with margins of 2–3 mm, rather conservative constraints were applied for CST avoidance in accordance with recent data, demonstrating impaired motor function after fractionated RT dose > 20–30 Gy.

Another potential cause for RT-associated motor complications is radiation necrosis (RN). Treatment volume, fractional dose, and total RT dose are risk factors for RN [78,79]. In postoperative stereotactic RT of BM, rates are up to 33.5% for predominantly large lesions [80]. In general, adjuvant hFSRT seems favourable over single-fraction radiosurgery, in terms of higher LC and lower incidences of RN, most likely due to breaks between fractions and a higher biologic effective dose (BED) [26,64,81,82,83,84]. In our cohort there was one case with symptomatic radiation necrosis, but symptoms subsided on anti-angiogenic medication (bevacizumab). There were no major motor complications after hFSRT observed. However, in the retrospective setting, there were no data on advanced motor assessment available to detect minor changes in motor strength that could still affect quality of life.

One limitation of our study is the yet unproven clinical relevance of sparing cortical and subcortical motor structures in radiotherapy. However, there are in-vitro data demonstrating RT mediated injury of white matter tracts and cortical atrophy. Furthermore, one study demonstrated a decline of fine motor skills after normo-fractionated RT in brain tumor patients, and another trial has shown that avoidance of the CST in high-single dose RT for AVM can preserve motor function. In our study, we could not demonstrate a correlation of motor skills with dose to nTMS-based DTI-FT due to the small number of patients and the retrospective character with no advanced neurologic assessment.

Compared to adjuvant RT of resected glioma for BM, there is no standard of care in terms of fractionation and dose, target delineation (+/− surgical tract), and additional margins to cover suspected infiltration zones so far. Furthermore, the cavity volume can vary over time after surgery. Several studies have demonstrated variable changes of the cavity volume within weeks after surgery [85,86,87]. Thus, this limits the results of our study, because we applied a postoperative stereotactic RT concept for resected BM, that is not standardized yet. Further, in this study BM amenable for surgery were rather large and symptomatic due to proximity to eloquent structures, perilesional edema, and a shift of the pyramidal tract, due to space-occupying effects. However, only preoperative DTI-FT_mot.TMS_ were available, which were integrated into our RT planning system and merged with postoperative planning CT and MRI. Secondary to regressive perilesional edema, a lower space-occupying shift of the CST and potential cavity dynamics in the early postoperative course, not all pre-operative DTI-FT_mot.TMS_ could be reliably fused with postoperative imaging. We addressed this issue by applying EF to keep imbalances minor and to enhance accuracy for fusion.

## 5. Conclusions

This study integrated functional subcortical fiber tracts into adjuvant hFSRT planning for the management of BM, demonstrating a proof-of-concept design for CST avoidance. Life expectancy will further increase in the future due to targeted and more effective systemic treatments among cancer patients. Thus, preserving a patient´s quality of life and self-independence will be of growing concern, emphasizing the need for tools to potentially reduce long-term side effects of current treatments. Further studies are warranted to evaluate the most suitable RT techniques for sparing of the cortical and subcortical motor structures. The real impact on clinical outcome needs to be investigated in prospective clinical trials.

## Figures and Tables

**Figure 1 cancers-15-00282-f001:**
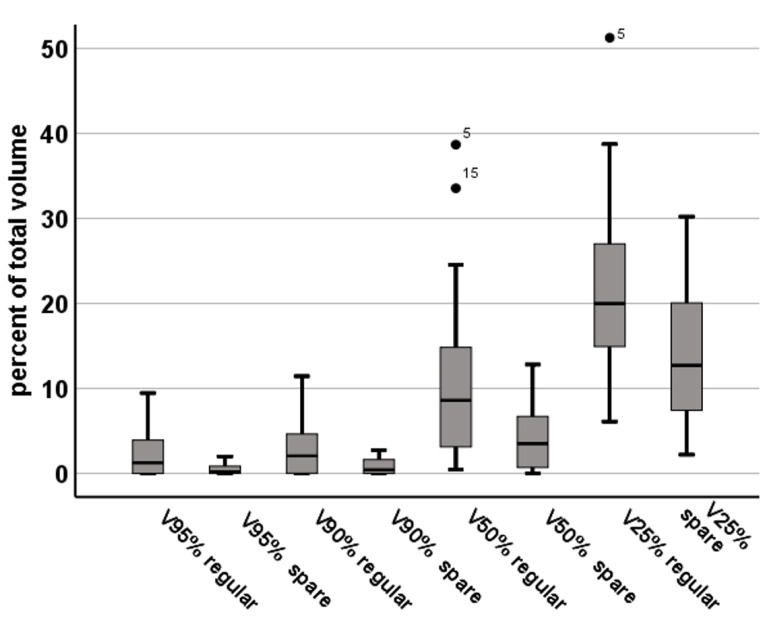
RT plan adaption for avoidance of FT_mot.tms_ translates into decline of proportional overlap, with corresponding isodose levels expressed as V95%, V90%, V50%, and V25% (%), with and without sparing (spare vs. regular). Mean V90% decreased by 74.2% (3.1% to 0.8%) and V50% by 66.8% (10.9% to 4.3%). All results are significant (*p* < 0.001).

**Figure 2 cancers-15-00282-f002:**
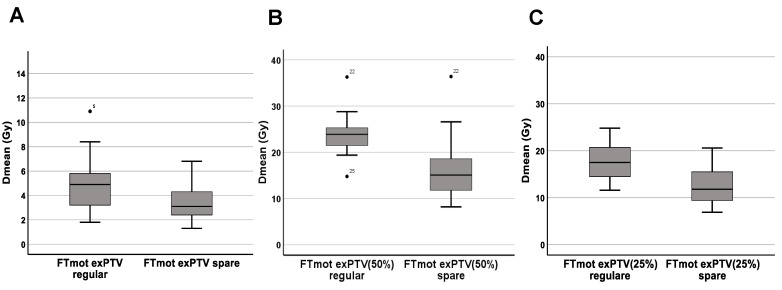
(**A**–**C**). Boxplots representing mean dose (Dmean) of different segments of DTI-FTnTMS based cortico-spinal tract beyond the planning target volume (FTmot exPTV), during regular treatment (FTmotexPTV regular), and after plan optimization for dose reduction (FTmotexPTV spare). Decline of dose exposition for all FTmotexPTV is minus 1.4 Gy (*p* < 0.001) (**A**). For segments both beyond the PTV and within the 17.5 Gy—isodose level (FTmotexPTV50%) Dmean reduction is minus 33.0% (range, 5.9–57.6%) from 23.4 Gy to 15.9 Gy (*p* < 0.001) (**B**). Looking at FTmot.nTMS portions both outside the PTV and within the 8.25 Gy—isodose level (FTmotexPTV25%) only, Dmean is decreased by (−29.2 ± 12.2%) (*p* < 0.001) (**C**).

**Figure 3 cancers-15-00282-f003:**
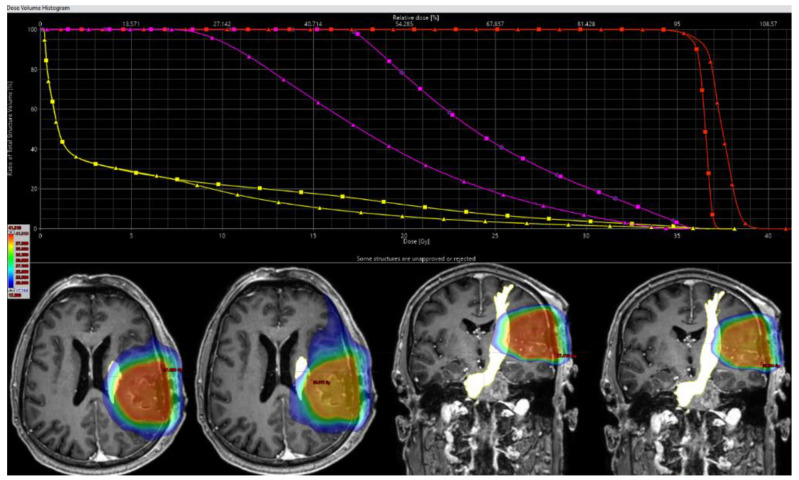
This figure illustrates the external beam plans for radiation therapy (RT) planning in one patient with a metastasis affecting the parietal lobe. The nTMS based DT-FT are seen delineated as organ at risk (FT_mot.tms_) in yellow and PTV in red. Of note, there is a slight overlap of the PTV with FT_mot.tms_. Dose distribution is portrayed in dose color wash from high (red) to lower (green to blue) dose. Each of the left planar views show regular treatment planning without taking FTmot into account. On both the left axial and coronal slide dose distribution is shown for regular plans, each of the right planes depicts dose for optimized plans. The higher dose is drawn away from the fiber tract. On a dose-volume histogram (DVH), dose reduction is seen as a steeper gradient for FT_mot.tms_ outside the PTV (FT_mot_exPTV) after optimization (yellow line, triangle), compared to the original RT treatment plan (yellow line, quadrate). Accordingly, the FT_mot.tms_ areas beyond the PTV and within the 50% isodose line (FT_mot_exPTV_50%_) are displayed in magenta. DVH for PTV is equal up to a relative dose of 95%, but with a higher maximum dose (red lines).

**Table 1 cancers-15-00282-t001:** Patient and tumor characteristics of 24 patients harboring metastases of different primary tumors close to motor-eloquent brain structures. All patients received preoperative navigated transcranial magnetic stimulation (nTMS)-based diffusion tensor imaging fiber tracking (DTI-FT_mot.tms_) of the corticospinal tract (CST). After resection they were treated with radiotherapy in an adjuvant setting. Motor deficits were graded according to the British Medical Research Council (BMRC) scale 0–5, with 0/5 = no contraction to 5/5 = full strength. Tumor entity was defined on histopathological. Abbreviations: CUP = Cancer of unknown primary, GTR = gross total resection, STR = subtotal resection.

**Total Number of Patients**			24
**Gender**	Female		17
	Male		7
**Age at primary treatment** **(Mean and range)**			60.4 years (32–78 years)
**Primary Tumor** **(Number of patients)**	Non-small cell lung cancer		7
	Breast cancer		5
	Colorectal carcinoma		2
	Ovarian cancer		2
	Ewing sarcoma		1
	Malignant melanoma		1
	Hidradenocarcinoma		1
	Urothelial carcinoma		1
	Vulvar cancer		1
	Hepatocellular carcinoma		1
	Gastric carcinoma		1
	Adenocarcinoma-CUP		1
**Tumor-affected hemisphere**	L		16
	R		8
**Tumor localization**	Frontal		13
	Temporal		2
	Parietal		9
**Extent of resection**	>95% = GTR		21
	80–95% = STR		3
**Preoperative systemic therapy (number of patients)**	Systemic therapy (total)		16
	Chemotherapy		14
	Immunotherapy		7
**Postoperative systemic therapy (number of patients)**	Systemic therapy (total)		18
	Chemotherapy		15
	Immunotherapy		12
	unknown		3
**Maximum tumor volume (mean and range)**			11.4 cm^3^ (0.4–57.1 cm^3^)
**Distance tumor—nTMS fibre tract (mean and range)**			0.4 cm (0.0–1.9 cm)
**Preoperative motor deficits**	BMRC	5/5	14
		4/5	7
		<3/5	3
**Postoperative motor deficits**	BMRC	5/5	13
		4/5	8
		<3/5	3
**Follow-up motor deficits (6 months after surgery)**	BMRC	5/5	17
		4/5	4
		<3/5	1
		unknown	2
**Motor deficits at follow-up before tumor progression (number of patients)**	n = 2		
	BMRC	5/5	2
		<3/5	1
**Motor deficits at maximum follow-up (number of patients)**	BMRC	5/5	20
		4/5	3
		<3/5	1

**Table 2 cancers-15-00282-t002:** Dose exhibition of nTMS derived fiber tract in regular plans (regular), and plans with dose adaption for sparing (spare). Mean dose of the total fiber tract decreased by 1.4 Gy. Segments of the CST beyond the PTV and inside 50%-Isodose-level reduced from 23.4 Gy to 15.9 Gy. Spatial overlap in percent of total volume for different isodose levels and FT_mot.tms_ were described as V100%_35Gy_, V95%_33.23Gy_, V90%_31.5Gy_, V50%_17.5Gy_, and V25%_8.78Gy_. After plan adaption volumes were significantly lower (*p* < 0.001).

Structure	Unit	Regular	Spare	*p*-Value
**nTMS.motor**	Dmean (Gy)	5.2 ± 2.4	3.8 ± 1.8	*p* < 0.001
**exPTV**	Dmean (Gy)	4.8 ± 2.1	3.4 ± 1.4	*p* < 0.001
**FT_mot_exPTV_50%_**	Dmean (Gy)	23.4 ± 3.3	15.9 ± 4.7	*p* < 0.001
**FT_mot_exPTV_25%_**	Dmean (Gy)	17.6 ± 4.2	12.5 ± 3.6	*p* < 0.001
**V100%_35Gy_**	(in %)	1.7 ± 2.3	0.2 ± 0.4	*p* < 0.001
**V95%_33.23Gy_**	(in %)	2.4 ± 3.1	0.5 ± 0.7	*p* < 0.001
**V90%_31.5Gy_**	(in %)	3.1 ± 3.7	0.8 ± 1.0	*p* < 0.001
**V50%_17.5Gy_**	(in %)	10.9 ± 9.8	4.3 ± 4.0	*p* < 0.001
**V25%_8.78Gy_**	(in %)	22.0 ± 10.5	14.2 ± 7.9	*p* < 0.001

**Table 3 cancers-15-00282-t003:** Dosimetric values of PTV in regular treatment plans (regular) and with sparing (spare) DTI-FT based CST. In comparison, Dmean of the PTV is not negatively affected. Coverage of target volume is equal after plan optimization. Changes of prescription isodose volume (PIV) were observed which decreased Selectivity and Paddick CI after recalculation, however were still appropriate. Gradient indices changes are minor, even though significant.

PTV	Regulare	DTI-FT Spared	*p*-Value
**Dmax (in Gy)**	37.2 ± 0.9	38.6 ± 1.6	*p* = 0.00
**Dmean (in Gy)**	35.6 ± 0.9	36.0 ± 1.2	*p* = 0.06
**Dmin (in Gy)**	32.8 ± 1.3	31.9 ± 1.6	*p* = 0.34
**Coverage**	1.0 ± 0.0	1.0 ± 0.0	*p* = 0.08
**Selectivitiy**	0.9 ± 0.1	0.8 ± 0.1	*p* = 0.00
**Paddick CI**	0.9 ± 0.1	0.8 ± 0.1	*p* = 0.00

**Table 4 cancers-15-00282-t004:** Dose exhibition of brain and organs at risk. In comparison there were some minor changes but after recalculation dose limits were kept respectively.

OAR	Side	DVH Value	Regular	DTI-FT Spared	*p*-Value
**Brain**		Dmean (Gy)	3.9 ± 1.9	3.5 ± 1.6	*p* = 0.51
**Healthy Brain (Brain minus Cavity)**		V28Gy (cc)	25.3 ± 14.9	25.9 ± 15.9	*p* = 0.40
		V24Gy (cc)	33.5 ± 19.7	35.0 ± 21.9	*p* = 0.22
**Hippocampus**	ipsilateral	Dmean (Gy)	1.8 ± 3.6	1.9 ± 3.7	*p* = 0.84
	contralateral	Dmean(Gy)	0.9 ± 1.8	0.9 ± 1.8	*p* = 0.91
**Brainstem**		Dmax (Gy)	1.7 ± 3.0	1.8 ± 3.2	*p* = 0.59
**Chiasm**		Dmax (Gy)	1.3 ± 2.0	1.2 ± 1.8	*p* = 0.19
**Optic Nerve**	ipsilateral	Dmax (Gy)	0.9 ± 1.4	0.9 ± 1.3	*p* = 0.75
	contralateral	Dmax (Gy)	0.7 ± 1.1	0.7 ± 1.1	*p* = 0.81
**Pituitary**		Dmax (Gy)	0.6 ± 1.0	0.6 ± 0.9	*p* = 0.41
**Lacrimalis Gland**	ipsilateral	Dmax (Gy)	1.3 ± 2.3	1.6 ± 3.0	*p* = 0.13
	contralateral	Dmax (Gy)	0.9 ± 1.5	0.9 ± 1.6	*p* = 0.99
**Cochlea**	ipsilateral	Dmax (Gy)	0.4 ± 0.6	0.4 ± 0.7	*p* = 0.10
	contralateral	Dmax (Gy)	0.1 ± 0.1	0.2 ± 0.1	*p* = 0.61
**Bulbus**	ipsilateral	Dmax (Gy)	1.3 ± 2.1	1.8 ± 3.1	*p* = 0.15
	contralateral	Dmax (Gy)	1.1 ± 1.7	1.0 ± 1.5	*p* = 0.29
**Lens**	ipsilateral	Dmax (Gy)	0.7 ± 1.3	0.8 ± 1.5	*p* = 0.85
	contralateral	Dmax (Gy)	0.5 ± 0.7	0.5 ± 0.7	*p* = 0.75

## Data Availability

The data presented in this study are available on reasonable request from the corresponding author.

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
