# Peer review of "Dose Reduction to Motor Structures in Adjuvant Fractionated Stereotactic Radiotherapy of Brain Metastases: nTMS-Derived DTI-Based Motor Fiber Tracking in Treatment Planning"

_cancers, 2022, doi:10.3390/cancers15010282_

Round 1

Reviewer 1 Report

Summary: In this manuscript, Diehl et al. describe the use of navigated transcranial magnetic stimulation (nTMS) in combination with DTI-based fiber tracking (DTI-FTmot. TMS) to reduce radiation dose to the motor cortex and motor tracts in a series of 24 patients undergoing radiation therapy for brain metastases. The authors demonstrate a reduction in mean radiation dose to the at-risk areas, without a reduction in the effective treatment dose. This is an interesting finding that could have value clinically. However, the authors fail to demonstrate that the magnitude of dose reduction they observed (33%) would be expected to reduce damage to the at-risk areas and resulting motor symptoms. The manuscript contains many technical details that are not adequately explained. Some references are old. Overall organization and presentation of results needs to be improved.

Specific comments:

11)     It is not entirely clear if the patients actually received the modified radiation plans incorporating DTI-FTmot. TMS, or if the modified plans were compared theoretically to the regular plans the patients actually received. My impression is the latter. Please make this very clear.

22)     In my opinion, the biggest issue with this study is correlation of dose reduction with clinically meaningful benefit. The authors show a statistically significant reduction in dose, but the magnitude of this dose reduction needs to be sufficient, at least in theory, to reduce motor symptoms. In the absence of actual data from patients, the authors would need to demonstrate rigorously, from previous literature, that a dose reduction of 33% from baseline would be expected to cause less tissue damage and less severe motor symptoms. Correlation with in vitro or in vivo data, or previous studies on radiation-associated motor symptoms in patients could be helpful.

33)     The manuscript contains a lot of technical detail. The authors should keep in mind that the target audience for this journal consists not only of radiation oncologists, but cancer researchers and clinicians from many different specialties and with different knowledge bases. The authors need to be able to explain the study to non-radiation oncologists. All abbreviations need to be explained. The dose conformity parameters and formulas (starting at line 170) may be more appropriate in the supplemental materials.

44)     Introduction, lines 48-50, the references on the incidence of brain metastases are 10+ years old.

55)     Introduction, lines 72-73: the statement that the precentral gyrus and corticospinal tracts are “not routinely respected in RT planning” needs to be supported by multiple high-quality references. Reference 24 is an atlas of at-risk intracranial regions published by a single university.

66)     Discussion, lines 350-372: do not simply re-state the results in the discussion section. Summarize them and explain their significance.

77)     Discussion, lines 412-427: this clinical outcome data needs to be included in the results section.

Author Response

Dear Editor,

Dear Reviewers,

We do appreciate immensely, that our manuscript “Dose reduction to motor structures in adjuvant fractionated stereotactic radiotherapy of brain metastases: nTMS-derived DTI-based motor fiber tracking in treatment planning“ (cancers-2053448) has been considered for re-submission. Thank you for your helpful comments. We have put great efforts into the major revision of our manuscript to thoroughly address the reviewers’ comments and questions to full satisfaction.

Thank you again for your consideration. We hope to hear back from you soon.

Yours sincerely,

Christian Diehl

Comments of Reviewer #1:

Initial Comment:

Summary: In this manuscript, Diehl et al. describe the use of navigated transcranial magnetic stimulation (nTMS) in combination with DTI-based fiber tracking (DTI-FTmot. TMS) to reduce radiation dose to the motor cortex and motor tracts in a series of 24 patients undergoing radiation therapy for brain metastases. The authors demonstrate a reduction in mean radiation dose to the at-risk areas, without a reduction in the effective treatment dose. This is an interesting finding that could have value clinically. However, the authors fail to demonstrate that the magnitude of dose reduction they observed (33%) would be expected to reduce damage to the at-risk areas and resulting motor symptoms. The manuscript contains many technical details that are not adequately explained. Some references are old. Overall organization and presentation of results needs to be improved.

Answer to the Initial Comment:

We would like to thank the reviewer for the constructive and helpful comments on our manuscript, the detailed assessment of our manuscript and for the critical evaluation. To regard the reviewer’s comment about the not yet clarified real benefit for this approach we would like to emphasize that according clinical trials are under way. Moreover, we provide detailed answers to the comments below and hope that our modifications made to the manuscript address the reviewer’s concerns adequately.

Comment #11:

It is not entirely clear if the patients actually received the modified radiation plans incorporating DTI-FTmot. TMS, or if the modified plans were compared theoretically to the regular plans the patients actually received. My impression is the latter. Please make this very clear.

Answer to Comment #11:

We would like to thank the reviewer for comment #1, pointing out that for the reader it appears not completely clear if the modified plans were finally applied for patient´s treatment. In this study we retrospectively modified already applied plans for avoidance of motor structures. No patient presented in this study had received modified experimental plans with specific dose adaptions to cortex or CST.

We emphasized this issue in both the abstract and in the method section:

  • “Completed fractionated stereotactic intensity-modulated RT (IMRT) plans were retrospectively analyzed and adapted…”, line 33-34.
  • “Presurgical obtained MRI datasets with EF and DTI-FTmot.TMS results of 24 patients who had been previously undergone local adjuvant RT after BM resection were imported into the TPS (Eclipse, version 13.0; Varian Medical Systems, Palo Alto, CA, USA) for retrospective dosimetric adaptions…“, line 145-147.
  • In this retrospective analysis adapted plans were not used for real patient treatment. “, line 430-431.

Comment #22:

 In my opinion, the biggest issue with this study is correlation of dose reduction with clinically meaningful benefit. The authors show a statistically significant reduction in dose, but the magnitude of this dose reduction needs to be sufficient, at least in theory, to reduce motor symptoms. In the absence of actual data from patients, the authors would need to demonstrate rigorously, from previous literature, that a dose reduction of 33% from baseline would be expected to cause less tissue damage and less severe motor symptoms. Correlation with in vitro or in vivo data, or previous studies on radiation-associated motor symptoms in patients could be helpful.

Answer to Comment 22:

We appreciate the reviewer’s thoughts on the at least theoretical clinical impact of dose reduction to cortical and subcortical motor structures. We admit that there are no validated and established dose constraints for motor eloquent structures so far because clinical data are scarce.  RT induced damage of white matter tracts occur from doses higher than 20 Gy and 30 Gy after 9-11 months and 4-6 months, respectively[1]. In regard of dose dependent cortical atrophy, the average change of cortical thickness increases with increasing dose. 34.6 Gy were demonstrated as averaged break point dose with higher thinning per Gy beyond that dose[2]. A recent study demonstrated that doses higher than fractionated doses of >20-30 Gy causes RT mediated injury of motor related white matter tracts as measured with DTI-parameters (fractional anisotropy and mean diffusity) resulting in a decline of fine motor skills[3]. Another study cited has shown that avoidance of the CST in high-single dose RT for AVM can preserve motor function. We changed the text accordingly

  • So far, there are only few studies demonstrating the potential impact of radiation to motor structres on motor function. In one study DTI-FTmot has been applied in treatment planning of Gamma Knife (GK) stereotactic radiosurgery (SRS) of solid tumors for pyramidal tract sparing, and 74 patients (BM 50%, mean volume 28.9 ± 26.2 ml) have been divided in two groups: DTI-avoidance (CST Dmax <15Gy) vs. control goup [75]. After 6 months, the DTI-avoidance group had significantly lower complications rates (not specified by authors), higher performance scores (Karnofsky Performance Status), and higher muscle strength of the affected limb [75]. “ Lines 720-726.
  • We made additions and re-arrangements of the text from line …-….
  • The following statements were added and revised: “One limitation of our study is the yet unproven clinical relevance of sparing cortical and subcortical motor structures in radiotherapy. However, there are in-vitro data demonstrating RT mediated injury of white matter tracts and cortical atrophy. Furthermore, one study demonstrated a decline of fine motor skills after normo-fractionated RT in brain tumor patients, and another trial has shown that avoidance of the CST in high-single dose RT for AVM can preserve motor function. In our study, we couln´t demonstrate a correlation of motor skills with dose to nTMS-based DTI-FT due to the small number of patients and the retrospective character with no advanced neurologic assessment.“ Lines:

We hope that these modifications address the reviewer’s thoughts sufficiently.

Comment #33:

The manuscript contains a lot of technical detail. The authors should keep in mind that the target audience for this journal consists not only of radiation oncologists, but cancer researchers and clinicians from many different specialties and with different knowledge bases. The authors need to be able to explain the study to non-radiation oncologists. All abbreviations need to be explained. The dose conformity parameters and formulas (starting at line 170) may be more appropriate in the supplemental materials.

Answer to Comment #33:

We agree with your consideration that the target audience of CANCERS are at a significant scale not mainly radiation oncologists. So, we tried to make it more readable and added furth explanations. We think that a certain number of indices is crucial to demonstrate plan quality with plan-adaption, thus we are convinced that coverage, selectivity and one conformity indices strengthen the quality of the manuscript. Indeed, the gradient indices are of minor significance, so we added it to the supplemental section at the end of the manuscript. We hope that the modifications address the reviewer´s concerns appropriately.

Coment #44:

Introduction, lines 48-50, the references on the incidence of brain metastases are 10+ years old.

Answer to Comment #44:

Thank you very much for this suggestion to update to more recent literature. We added highly published reference on epidemiologic data for brain metastases[4-6] and hope this meets your expectations.

Comment #55:

Introduction, lines 72-73: the statement that the precentral gyrus and corticospinal tracts are “not routinely respected in RT planning” needs to be supported by multiple high-quality references. Reference 24 is an atlas of at-risk intracranial regions published by a single university.

Answer to Comment #55:

Thank you for this comment, indeed the statement referring to one single-institution statement appears vague. There are some high-quality references about contouring and potential risks of radiation, which do not address the need for contouring motor areas as organs at risk. Mainly secondary to missing clinical evidence to do so.

We accordingly rephrased that sentence and added more references as requested.

  • Routine sparing of the precentral gyrus and CST in RT planning is not supported so far, mainly because the real clinical impact is not sufficiently proven[7-11].

Comment #66:

do not simply re-state the results in the discussion section. Summarize them and explain their significance.

Answer to Comment #66:

We appreciate this comment, instead of just re-stating the results in the discussion section, we now rephrased this section for more data interpretation and discussion.

Comment #77:

Discussion, lines 412-427: this clinical outcome data needs to be included in the results section.

Answer to Comment #77:

Thank you for the advice. We agree with the reviewer that the outcome data should be mainly reported in the result section and being discussed within the discussion.

Closing comment from authors:

We would like to thank the reviewer for the constructive comments on our manuscript. We have the impression that incorporation of the points raised has strengthened our work.

References:

  1. Connor M, Karunamuni R, McDonald C, White N, Pettersson N, Moiseenko V, Seibert T, Marshall D, Cervino L, Bartsch H et al: Dose-dependent white matter damage after brain radiotherapy. Radiother Oncol 2016,121(2):209-216.
  2. Karunamuni R, Bartsch H, White NS, Moiseenko V, Carmona R, Marshall DC, Seibert TM, McDonald CR, Farid N, Krishnan A et al: Dose-Dependent Cortical Thinning After Partial Brain Irradiation in High-Grade Glioma. Int J Radiat Oncol Biol Phys 2016, 94(2):297-304.
  3. Salans M, Tibbs MD, Karunamuni R, Yip A, Huynh-Le MP, Macari AC, Reyes A, Tringale K, McDonald CR, Hattangadi-Gluth JA: Longitudinal change in fine motor skills after brain radiotherapy and in vivo imaging biomarkers associated with decline. Neuro Oncol 2021, 23(8):1393-1403.
  4. Cagney DN, Martin AM, Catalano PJ, Redig AJ, Lin NU, Lee EQ, Wen PY, Dunn IF, Bi WL, Weiss SE et al: Incidence and prognosis of patients with brain metastases at diagnosis of systemic malignancy: a population-based study. Neuro Oncol 2017, 19(11):1511-1521.
  5. Lamba N, Wen PY, Aizer AA: Epidemiology of brain metastases and leptomeningeal disease. Neuro Oncol 2021, 23(9):1447-1456.
  6. Sperduto PW, Mesko S, Li J, Cagney D, Aizer A, Lin NU, Nesbit E, Kruser TJ, Chan J, Braunstein S et al: Survival in Patients With Brain Metastases: Summary Report on the Updated Diagnosis-Specific Graded Prognostic Assessment and Definition of the Eligibility Quotient. J Clin Oncol 2020, 38(32):3773-3784.
  7. Kirkpatrick JP, Marks LB, Mayo CS, Lawrence YR, Bhandare N, Ryu S: Estimating normal tissue toxicity in radiosurgery of the CNS: application and limitations of QUANTEC. J Radiosurg SBRT 2011, 1(2):95-107.
  8. Milano MT, Grimm J, Niemierko A, Soltys SG, Moiseenko V, Redmond KJ, Yorke E, Sahgal A, Xue J, Mahadevan A et al: Single- and Multifraction Stereotactic Radiosurgery Dose/Volume Tolerances of the Brain. Int J Radiat Oncol Biol Phys 2021, 110(1):68-86.
  9. Niyazi M, Brada M, Chalmers AJ, Combs SE, Erridge SC, Fiorentino A, Grosu AL, Lagerwaard FJ, Minniti G, Mirimanoff RO et al: ESTRO-ACROP guideline "target delineation of glioblastomas". Radiother Oncol 2016, 118(1):35-42.
  10. Scoccianti S, Detti B, Gadda D, Greto D, Furfaro I, Meacci F, Simontacchi G, Di Brina L, Bonomo P, Giacomelli Iet al: Organs at risk in the brain and their dose-constraints in adults and in children: a radiation oncologist's guide for delineation in everyday practice. Radiother Oncol 2015, 114(2):230-238.
  11. Soliman H, Ruschin M, Angelov L, Brown PD, Chiang VLS, Kirkpatrick JP, Lo SS, Mahajan A, Oh KS, Sheehan JP et al: Consensus Contouring Guidelines for Postoperative Completely Resected Cavity Stereotactic Radiosurgery for Brain Metastases. Int J Radiat Oncol Biol Phys 2018, 100(2):436-442.

Reviewer 2 Report

The present manuscript is interesting because is important to avoid CST during RT treatment for BM to preserve the functionality of the brain and more studies are needed to improve the knowledge about that and improve the treatment for BM. However, there are several points during the study that must be reviewed:

First of all, why the mean volumes of the resection cavity based on planning MRI and FTmot.TMS are so different? There is some case to add a figure to an example of both masks and view the differences?

The Figure cited in sentence 220 is not the patient's case. The actual figure 1 is a boxplot from different patients and is correctly referenced in sentence 240. Please review the Figure in sentence 220 and change if it necessary. I suggest adding the patient case to illustrate the PTV overlapped with FT. Similarly, review figure 1 and table 2 referenced in sentence 241. Are they the correspondent to sentences 240-241?

Table 1, there is missing some acronyms: STR and GTR.

Have the authors compared the BMRC by a paired t-test? Have the authors studied if the patients with BMRC of 5/5 in preoperative motor deficits are the same in postoperative motor, and during the follow-ups? There is some fluctuation between the scales? On the other hand, I suggest adding information about the scale of BMRC: Which is the best and the worst score on the scale? There is any correlation between motor deficits and the total dose or the dose in the OARs?

Are there only 2 patients with tumour progression? Is there more information about the motor deficits at follow-up before tumour progression, for example, if they are worse on BMRC scale?

The authors analyzed the results by Kaplan-Mayer. I suggest adding the figure as a supplementary figure.

Finally, the authors are concluding that DTI-FT radiate less in the motor fibre tracts, however, there are no statistical differences between the dose of regular and spared DTI-FT in table 4. Have the authors segmented or grouped the patients between the distance of OARs and the dose received? There is any correlation studied about that? There is any correlation studied about that? This approach, of grouping the patients by distance, for example, might improve the differences between treatments. The results might be a good indication to perform the proposed RT for selected patients.

Author Response

Dear Editor,

Dear Reviewers,

We do appreciate immensely, that our manuscript “Dose reduction to motor structures in adjuvant fractionated stereotactic radiotherapy of brain metastases: nTMS-derived DTI-based motor fiber tracking in treatment planning“ (cancers-2053448) has been considered for re-submission. Thank you for your helpful comments. We have put great efforts into the major revision of our manuscript to thoroughly address the reviewers’ comments and questions to full satisfaction.

Thank you again for your consideration. We hope to hear back from you soon.

Yours sincerely,

Christian Diehl

Comments of Reviewer #1:

Initial Comment:

The present manuscript is interesting because is important to avoid CST during RT treatment for BM to preserve the functionality of the brain and more studies are needed to improve the knowledge about that and improve the treatment for BM. However, there are several points during the study that must be reviewed.

Answer to the Initial Comment:

We would like to thank the reviewer for the constructive and helpful comments on our manuscript, the detailed assessment of our manuscript and for the critical evaluation. We will provide detailed answers to the comments below and hope that our modifications made to the manuscript address the reviewer’s concerns adequately.

Comment #1:

First of all, why the mean volumes of the resection cavity based on planning MRI and FTmot.TMSare so different? There is some case to add a figure to an example of both masks and view the differences?

Answer to Comment #1:

Thank you very much for this comment. Indeed, this sentence tends to be misleading. We have tried to describe the volumes of two different structures: the volume of the resection cavity as measured on MRI and the volume of the nTMS-based motor structure (FTmot.TMS).

The sentence was re-phrased accordingly:

  • Mean volume of the resection cavity as measured on planning MRI was 7.1 ml ± 8.6 (range, 0.5 - 31 ml), the volume of FTmot.TMS was 54.6 ml ± 18.7 (range, 19.5 – 91.0 ml), respectively“)

 We hope that we could create more clarity this way.

Comment #22:

The Figure cited in sentence 220 is not the patient's case. The actual figure 1 is a boxplot from different patients and is correctly referenced in sentence 240. Please review the Figure in sentence 220 and change if it necessary. I suggest adding the patient case to illustrate the PTV overlapped with FT. Similarly, review figure 1 and table 2 referenced in sentence 241. Are they the correspondent to sentences 240-241?

Table 1, there is missing some acronyms: STR and GTR.

Answer to Comment 22:

Thank you very much for your notes. The sentence in line 240 wrongly pointing to figure 1 was deleted, because we refer to the example case correctly as figure 3 in line 247. We added a remark to the capture of figure 3 (exemplary case), that the PTV slightly overlapped with the FT. In line 241 table 1 (patient and tumor characteristics is correct). We added the missing acronyms as suggested.

We hope that these modifications address the reviewer’s thoughts sufficiently.

Comment #3:

Have the authors compared the BMRC by a paired t-test? Have the authors studied if the patients with BMRC of 5/5 in preoperative motor deficits are the same in postoperative motor, and during the follow-ups? There is some fluctuation between the scales? On the other hand, I suggest adding information about the scale of BMRC: Which is the best and the worst score on the scale? There is any correlation between motor deficits and the total dose or the dose in the OARs?

Answer to Comment #3:

We appreciate your comments on the data/result about BMRC. First of all, we introduced the scale in the method section explaining the best and worst score. Thanks to your comment, we re-analyzed the clinical data and found one error. Indeed, one patient had a new grade 4/5 motor deficit postoperatively that subsided over the course of disease. We made an appropriate correction in table one.  Since there was only one patient with transient aggravated motor deficit with improvement on bavecizumab after radiation therapy we dispensed a t-test for correlation of adverse change of motor strength and dose to motor structures.

Coment #4:

Are there only 2 patients with tumour progression? Is there more information about the motor deficits at follow-up before tumour progression, for example, if they are worse on BMRC scale?

Answer to Comment #44:

Thank you for this interesting note. We observed one patient with confirmed PD on PET, two more patients had suspected PD. We corrected table 1 accordingly.  The three patients with local tumor recurrence did not reveal new deficits, we added this information to the result section.

Comment #5:

The authors analyzed the results by Kaplan-Mayer. I suggest adding the figure as a supplementary figure.

Answer to Comment #5:

We did as proposed. When editing the Kalan-Meyer-curves we found one mistake: OS at 12 months is 68.1% (95%-CI 48.5 – 87.7%) instead of 73.4% (95%-CI 55.0 – 91.8%). We made a respective correction.

Comment #6:

Finally, the authors are concluding that DTI-FT radiate less in the motor fibre tracts, however, there are no statistical differences between the dose of regular and spared DTI-FT in table 4. Have the authors segmented or grouped the patients between the distance of OARs and the dose received? There is any correlation studied about that? There is any correlation studied about that? This approach, of grouping the patients by distance, for example, might improve the differences between treatments. The results might be a good indication to perform the proposed RT for selected patients.

Answer to Comment #6:

Thank you for the comment. We would like to point out, that the dose to the DTI-FT is displayed/summarized in Table 2. Table 4 is depicting all additional OAR, which are no motor structures. We agree with you that definitely not all patients will benefit from sparing motor structures, due to a long distance between PTV and OAR with normal dose fall of in between.  It seems obvious that the closer the distance between cavity and FT is, the higher the potential dose reduction will be. Since we don´t have the clinical data yet for the dose constraint regarding the FT, we couldn´t find a correlation between the distance between cavity and FT from which a plan adaption for FT for avoidance would be useful. It depends on several factors such as dose, fractionation and size of the PTV. We hope this addresses the reviewer´s thoughts sufficiently.  

Closing comment from authors:

We would like to thank the reviewer for the constructive comments on our manuscript. We have the impression that incorporation of the points raised has strengthened our work.

Reviewer 3 Report

Post-operative management of brain metastases (BM) located close to motor areas and tracts remains challenging due to potential damaging of these motor-eloquent brain structures in the adjuvant radiotherapy setting. In this specific setting, hypo-fractionated stereotactic radiotherapy (hFSRT) remains a highly effective modality for local control of resection cavity and prevention of recurrence due to minimal residual disease. In this proof-of-concept study, the authors integrated the pre-operative mapping of subcortical fiber tracts (i.e., the corticospinal tracts) fused with post-operative MRI data collected from 24 patients into the adjuvant hFSRT planning for the management of BM. Specifically, the authors took advantage of the availability of pre-operative data from navigated transcranial magnetic stimulation (nTMS) motor mapping combined with diffusion tensor imaging (DTI)-based fiber tracking (DTI-FTmot.TMS) of the CST which were fused, by applying elastic image fusion techniques, with co-registered post-operative MRI data to correct and compensate for brain shifts after tumor resection. This approach led to a more accurate target delineation with respect to motor structures—defined as the minimum lesion-to-CST distance—for radiotherapy planning. Based on these analyses, the authors show that this approach could be very useful for dosimetric decisions and could lead to important reductions in total radiotherapy dosages while achieving  motor structure sparing with no negative consequences on the planned total volume (PTV) coverage and conformity. 

The study is well written and provides a clear presentation of the background, data and findings. Importantly, the authors discussed the limitations of their study including its retrospective basis, the lack of advanced motor assessment data, the lack of post-operative DTI-FTmot.TMS data, etc. While a prospective clinical trial is ultimately needed to validate this approach, the retrospective data analyzed here, and the general approach behind the study, represent valuable tool which should be of great interest as proof-of-concept and foundational. With the expected increases in life expectancy for these BM patients, finding new strategies aimed at sparing the motor brain structures of the patients in the adjuvant radiotherapy setting, with important consequences in terms of patient quality of life and the preservation of self-independence, remains a critically important and unmet medical need. 

Author Response

Dear Editor,

Dear Reviewers,

We do appreciate immensely, that our manuscript “Dose reduction to motor structures in adjuvant fractionated stereotactic radiotherapy of brain metastases: nTMS-derived DTI-based motor fiber tracking in treatment planning“ (cancers-2053448) has been considered for re-submission. Thank you for your helpful comments. We have put great efforts into the major revision of our manuscript to thoroughly address the reviewers’ comments and questions to full satisfaction.

Thank you again for your consideration. We hope to hear back from you soon.

Yours sincerely,

Christian Diehl

Comments of Reviewer #3:

Post-operative management of brain metastases (BM) located close to motor areas and tracts remains challenging due to potential damaging of these motor-eloquent brain structures in the adjuvant radiotherapy setting. In this specific setting, hypo-fractionated stereotactic radiotherapy (hFSRT) remains a highly effective modality for local control of resection cavity and prevention of recurrence due to minimal residual disease. In this proof-of-concept study, the authors integrated the pre-operative mapping of subcortical fiber tracts (i.e., the corticospinal tracts) fused with post-operative MRI data collected from 24 patients into the adjuvant hFSRT planning for the management of BM. Specifically, the authors took advantage of the availability of pre-operative data from navigated transcranial magnetic stimulation (nTMS) motor mapping combined with diffusion tensor imaging (DTI)-based fiber tracking (DTI-FTmot.TMS) of the CST which were fused, by applying elastic image fusion techniques, with co-registered post-operative MRI data to correct and compensate for brain shifts after tumor resection. This approach led to a more accurate target delineation with respect to motor structures—defined as the minimum lesion-to-CST distance—for radiotherapy planning. Based on these analyses, the authors show that this approach could be very useful for dosimetric decisions and could lead to important reductions in total radiotherapy dosages while achieving  motor structure sparing with no negative consequences on the planned total volume (PTV) coverage and conformity. 

The study is well written and provides a clear presentation of the background, data and findings. Importantly, the authors discussed the limitations of their study including its retrospective basis, the lack of advanced motor assessment data, the lack of post-operative DTI-FTmot.TMS data, etc. While a prospective clinical trial is ultimately needed to validate this approach, the retrospective data analyzed here, and the general approach behind the study, represent valuable tool which should be of great interest as proof-of-concept and foundational. With the expected increases in life expectancy for these BM patients, finding new strategies aimed at sparing the motor brain structures of the patients in the adjuvant radiotherapy setting, with important consequences in terms of patient quality of life and the preservation of self-independence, remains a critically important and unmet medical need. 

Answer to the Reviewer´s:

Thank you very much for the positive over all response.

Round 2

Reviewer 1 Report

Overall, the authors have successfully applied feedback and answered comments more or less satisfactorily. They have incorporated more literature and used it to better support their conclusions.